# Image-Based Crack Detection Method for FPSO Module Support

**Xin Su [1], Ziguang Jia [1], Guangda Ma [1], Chunxu Qu [2],\*, Tongtong Dai [2] and Liang Ren [2]**

[1] School of Ocean Science and Technology, Dalian University of Technology, Dalian 124221, China; 793962321@mail.dlut.edu.cn (X.S.); jiaziguang@dlut.edu.cn (Z.J.); mgd18642278277@mail.dlut.edu.cn (G.M.)
[2] Faculty of Infrastructure Engineering, Dalian University of Technology, Dalian 116024, China; tongtongdai@mail.dlut.edu.cn (T.D.); renliang@dlut.edu.cn (L.R.)
\* Correspondence: quchunxu@dlut.edu.cn

**Abstract:** Floating Production Storage and Offloading (FPSO) is essential offshore equipment for developing offshore oil and gas. Due to the complex sea conditions, FPSOs will be subjected to long-term alternate loads under some circumstances. Thus, it is inevitable that small cracks occur in the upper part of the module pier. Those cracks may influence the structure's safety evaluation. Therefore, this paper proposes a method for the FPSO module to support crack identification based on the PSPNet model. The main idea is to introduce an attention mechanism into the model with Mobilenetv2 as the backbone of the PSPNet, which can fuse multiple feature maps and increase context information. The detail feature loss caused by multiple convolutions and compressions in the original model was solved by applying the proposed method. Moreover, the attention mechanism is introduced to enhance the extraction of adequate information and suppress invalid information. The mPA value and MIoU value of the improved model increased by 2.4% and 1.8%, respectively, through verification on FPSO datasets.

**Keywords:** FPSO; deep learning; semantic segmentation; MobileNetv2; PSPNet

## 1. Introduction

A crack is a common form of structural damage that will lead to a decrease in the overall strength of the structure, with stress concentrating at both ends of the crack. A welding crack is a common defect in welding parts specifically. Under the joint action of welding stress and other brittle factors, the bonding force of metal atoms in the local area of the welded joint is destroyed and a new interface is formed by the gap. Generally, small cracks will not cause safety hazards in a short period. Yet, if the equipment is exposed to cracks in harsh conditions for a long time, cracks may expand and cause structural damage. This paper focuses on Floating Production Storage and Offloading (FPSO) cracks, which are specifically welding cracks. Those occur in the welding process and have a certain incubation period. With the discovery of large numbers of deep-sea, associated, and marginal oil and gas fields, and as we move away from fixed oil and gas exploration platforms, FPSO cracks are becoming increasingly prominent thanks to the low investment, short construction periods, and reusable materials applied to harvest the new fields. As such, FPSO cracks are coming to play an increasingly important role in the field of offshore engineering. When working in the offshore alternating load environment, the fatigue life of the upper pier module directly affects the service life of FPSO [1].

According to the results of a recent 5-year special comprehensive inspection of the upper pier module of FPSO, 44 of the 84 support piers of a ship were found to have cracks in the weld joints. Cracks were found in 71 out of 336 welds connected to the main deck, accounting for 21.1% of the total number of welds.

When FPSOs are subjected to wave action, the main load is created by the vertical shear force, torsional moment, and vertical bending moment. The wave causes buoyancy to change along the length of the ship, resulting in a total longitudinal moment, in which

case, there will be mid-arch and mid-sag bends along the length of the ship. When the wavelength is equal to or close to the length, the total longitudinal moment of the wave is most significant, and the total longitudinal strength of the ship hull of FPSO is determined by the sagging state. According to the statistics on marine accidents in ship and ocean engineering, structural fatigue damage is the main cause of marine accidents. Seventy percent of the damage to large ships with captains over 200 m is fatigue damage. With fatigue damage to the hull structure, the focus is on the mutation area and stress concentration of the connection part of the component, namely, the fatigue of the connection node and high-stress area of the structure. FPSO is subjected to alternating stresses of the wave load for its whole life. In cases of strong wind and waves, cracks have been produced at the weld joint between the upper module support pier and the main deck. In the case of arch bends, the deck may be torn, resulting in an oil tank leakage accident. Figure 1 shows a crack distribution diagram for the upper module support pier.

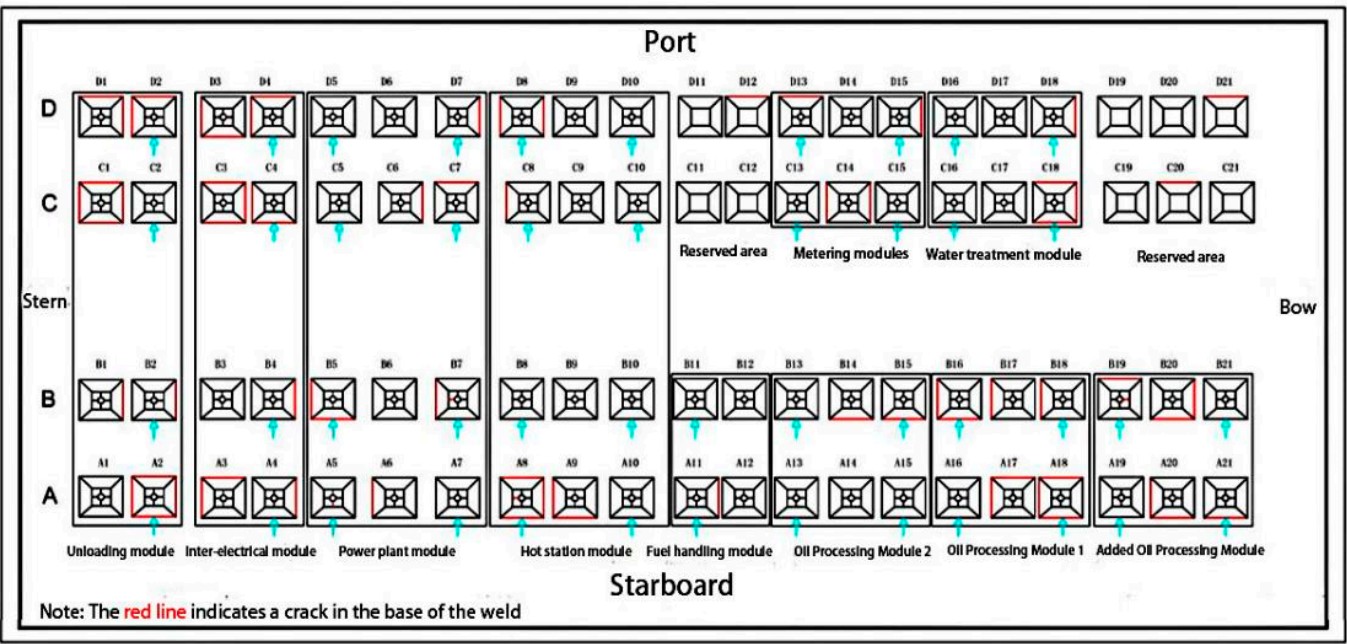

**Figure 1.** Crack distribution diagram of upper module support pier. (A–D represents the line number).

FPSO is an important part of the offshore oil industry. Figure 2 shows a diagram of a crack in the upper module. If manual inspection is to be used to detect cracks, this usually requires the cooperation of engineers and technicians, project management personnel, ocean engineering experts, safety experts, and other relevant personnel. When human eyes detect cracks, it is easy to introduce subjective factors, and there is a residual phenomenon, which is inefficient. Since neural networks have undergone many innovations in recent years, a crack-recognition method based on deep learning has come to be widely used, providing an efficient method for structural health detection.

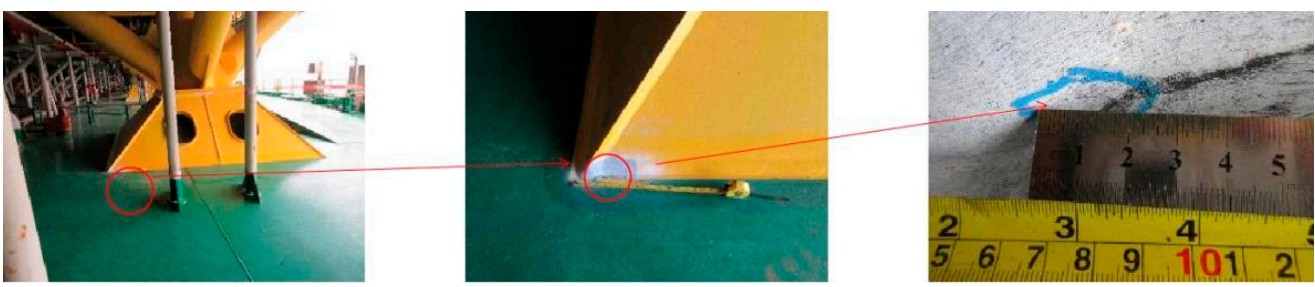

**Figure 2.** Diagram of the crack in the upper module.

## 2. Related Work

In early research, computer vision was applied to automatically detect cracks through image processing. Image processing [2] mainly used edge detection and segmentation technology. Cheng et al. [3] proposed a threshold segmentation algorithm based on reducing the sample space. Song et al. [4] proposed a method to detect surface cracks based on the adaptive Canny algorithm and iterative threshold segmentation algorithm.

With the continuous development of computer science, machine learning methods are gradually coming to be applied to image processing. In 2012, Hinton's team participated in the ImageNet image-recognition competition and won the championship by establishing AlexNet [5], proving the potential of deep learning and its Convolutional Neural Network (CNN). Since then, CNN has entered the industrial world. Farooq et al. [6] carried out experiments with different material properties and loading conditions and then simulated a healthy structure and two damaged structures with one and two small cracks, respectively. For binary classification of structures with or without damage, the prediction accuracies of neural networks and support vector machines were 93.2% and 96.66%, respectively. The average prediction accuracies of neural networks and support vector machines were 83.5% and 90.05%, respectively, for multi-class classification problems. In 2014, Ross et al. [7] improved the CNN algorithm, used it to generate candidate regions for the image, used the deep convolution network to extract features for each region, and applied the SVM algorithm to classify and judge the features. Finally, the R-CNN algorithm was proposed, which greatly improved the efficiency of target detection. The algorithm [8] only focuses on important information and ignores the rest. In 2015, the Visual Geometry Group of Oxford University [9] proposed the VGG algorithm, which improved the depth and accuracy of neural networks by using $3 \times 3$ convolution kernels. In the same year, Redmon proposed [10] the YOLO algorithm; it is different from the R-CNN algorithm-CNN algorithm, which has high accuracy but a slow speed, while the YOLO algorithm sacrifices accuracy for the detection speed. Recently, the fifth-generation YOLO algorithm has achieved a balance between speed and accuracy. The ResNet proposed by He [11] makes the existence of ultra-high-depth neural networks possible. It is based on the VGG network, and the algorithm adds residual cells for modification through a short circuit mechanism, as well as adding a short circuit connection between every two layers to enable residual learning. At present, most mature solutions in various fields of artificial intelligence are based on the VGG, YOLO, and R-CNN algorithms. In 2017, through further innovation of the ResNet algorithm, the DenseNet algorithm [12] was born. Then came the lightweight neural network based on mobile devices. Transformer [13] also came into being. It abandoned the traditional CNN and RNN; in place of those, the entire network structure is composed of attention mechanisms. These new algorithms, updated with the latest developments of the times, have opened up a new world in the field of deep learning.

Sometimes object detection does not apply to certain objects in a scene, meaning another method of target recognition is required, that is, semantic segmentation. Chorowski et al. [14] introduced the FCN full-convolution network to image crack detection and proposed a crack FCN model that effectively reduced the error of crack detection against complex backgrounds. In the research by Hamishebahar et al. [15], the Att-Unet model was proposed. The model is improved based on the attention mechanism of a full-convolution neural network, and it can realize end-to-end pixel-level crack segmentation. Zheng et al. [16] proposed an algorithm based on SegNet and separable convolution of bottleneck depth with residuals, which can be used for high-precision light bridge concrete crack detection. Kang and Cha [17] proposed a new network called STRNet for real-time segmentation of pixel-level cracks in complex scenes. Li et al. [18] proposed a crack-detection method consisting of two stages, which can support effective detection based on large-scale collected images. The first stage is crack recognition, in which crack images are recognized by finetuning the VGG16 model. In the second stage, UNet++ is used as a model [19] for pixel-level crack semantic segmentation of images. Fan et al. [20]

proposed a new depth residual convolution neural network to establish a high-performance pavement crack-detection system.

In the field of structural health detection, the CNN model has also been fully applied. Chen et al. [21] proposed high-dimensional phase space reconstruction with a CNN model, converting the high-dimensional reconstructed attractors to images and feeding them into a CNN model. Pathak [22] proposed using CNN to monitor ridge health, and encoding methods are promising for future studies of CNN-based seismic damage evaluation. Zhao et al. [23] used a conventional and metaheuristic-tuned artificial neural network to predict the compressive strength of manufactured-sand concrete.

## 3. Methods

### 3.1. Baseline PSPNet Model

The PSPNet model is based on the FCN model and adopts a residual network to extract features. This model puts forward a Pyramid Scene Parsing module, which has been greatly improved for image segmentation. The network used in this paper is a modified feature-extraction network based on the PSPNet model. The original ResNet was composed of conv1_x, conv2_x, conv3_x, conv4_x, and conv5_x, among which conv1_1, conv2_1, conv3_1, conv4_1, and conv5_1 were downsampled once, respectively, giving a total of five instances of downsampling. The output feature map is $1/32$ of the input size. In PSPNet, the ResNet module applies a dilated convolution, where conv1_x, conv2_x, and conv3_x remain unchanged. The stride of the first $1 \times 1$ convolution in conv4_1 is changed from 2 to 1, and the $3 \times 3$ convolution in conv4_x adopts dilated convolution with sampling rates of 2. The stride of the first $1 \times 1$ convolution in conv5_1 is changed from 2 to 1, and the $3 \times 3$ convolution in conv5_x adopted an expansion convolution with a sampling rate of 4. In this way, the downsampling times of the ResNet module when applying the expansion convolution are reduced by two to three times, and the output characteristic graph is one-eighth the input size; at the same time, the sensibility field of the feature map is enlarged by expanding the convolution.

### 3.2. Comparison of MobileNet and ResNet Structures

The main ideas of MobileNetv2 are deeply separable convolution, a linear bottleneck layer, and inverted residual [24]. Deeply separable convolution can greatly reduce the parameters of the model. Feature extraction and combination of the ordinary convolution layer are completed and outputted in one, while deeply separable convolution first uses a $3 \times 3$ convolution kernel with a thickness of 1, then uses a $1 \times 1$ convolution kernel (pointwise convolution) to adjust the numbers of channels, to separate feature extraction from feature combination. An inverse residual structure is the projection convolution of modules connected with residuals to raise the dimension, followed by depthwise convolution, and finally, using projection convolution to reduce the dimension. In MobileNetv1, the parameters of the depth convolution kernel are mostly 0, which means that its convolution kernel does not play a role in feature extraction. Due to the greater numbers of input and output channels in depthwise convolution, more information can be extracted. Figure 3 shows a diagram of MobileNetv2; as can be seen, feature maps of different scales can be outputted from MobileNetv2.

MobileNetv2 is a lightweight network. It retains the residual module, but compared to the traditional residual network, the number of parameters is greatly reduced. Figure 4 presents a comparison of the MobileNet and ResNet structures.

The ResNet module uses standard convolution to extract features, which reduces the dimension, convolves, and then increases the dimension. The MobileNet module has a depthwise convolution function, so the process of extracting features is to increase the dimension, convolve, and then reduce the dimension. Visually, ResNet has an hourglass microstructure, while MobileNetv2 has a rotating-type microstructure. Accordingly, the structure of MobileNetv2 is called a residual block.

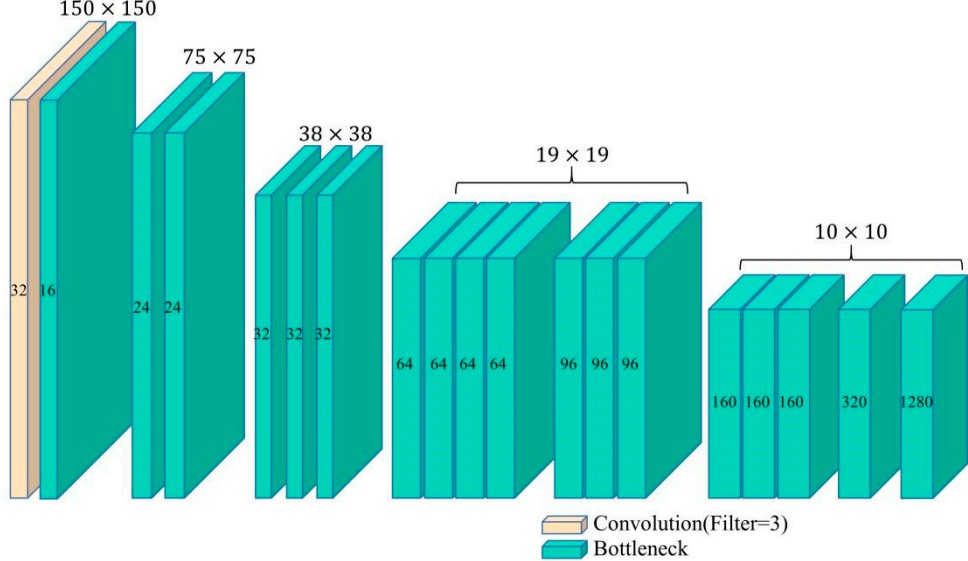

**Figure 3.** Structure diagram of MobileNetv2.

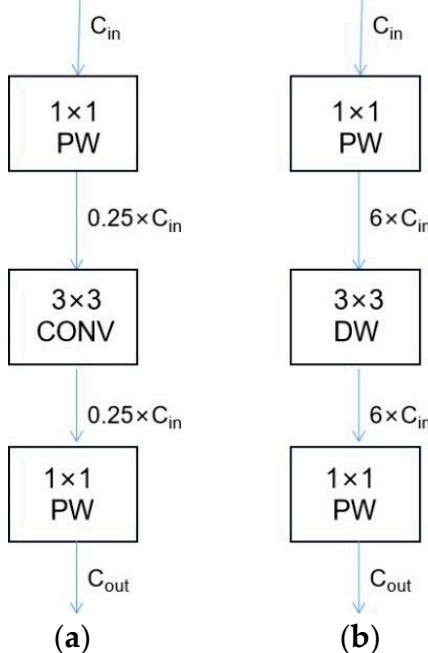

**Figure 4.** Comparison of MobileNet and ResNet structures: (**a**) ResNet; (**b**) MobileNetv2.

The important module in PSPNet is the Pyramid Scene Parsing module. The pyramid pooling module combines four features with different scales: $1 \times 1$, $2 \times 2$, $3 \times 3$, and $6 \times 6$. Features of different depths are pooled at different scales according to the input, and the feature dimension is reduced to one-quarter of the original through the $1 \times 1$ convolution layer. Finally, these pyramid features are directly upsampled to the same size as the input features, then combined with the input features to obtain the final output features. The process of feature merging involves fusing the detail feature (shallow feature) and global feature (deep feature) of the target. Figure 5 shows a structure diagram of PSPNet. Figure 6 shows a PSPNet structure diagram with Mobile as the backbone. The difference compared to the original PSPNet is that MobileNetv2 is used as the backbone, and the feature maps at six different scales generated by MobileNetv2 are used as the input.

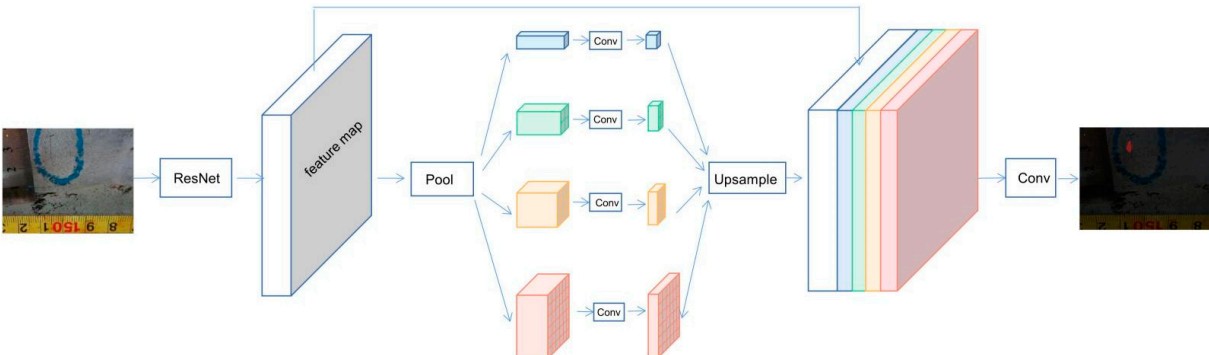

**Figure 5.** Structure diagram of PSPNet.

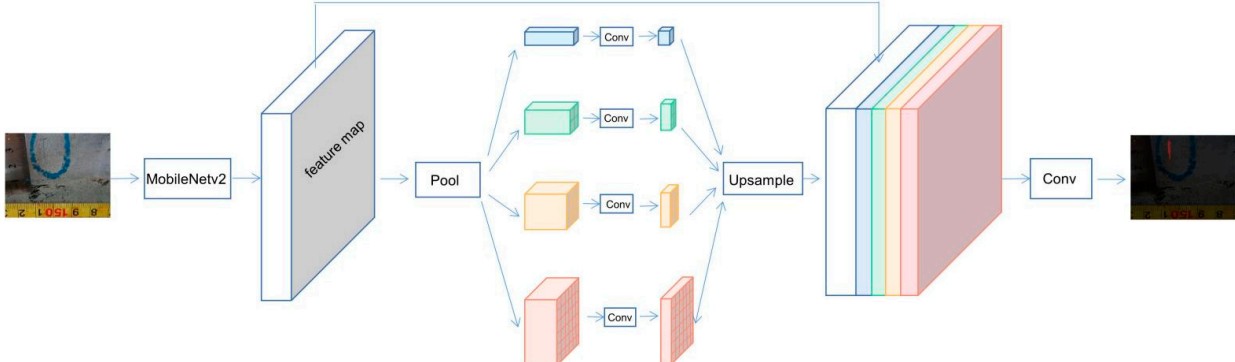

**Figure 6.** PSPNet structure diagram with Mobile as the backbone.

### *3.3. Improved PSPNet Model*

The main idea of an attention mechanism is to enhance useful information and suppress useless or minor information. This paper introduces an attention mechanism to the Mobilev2-SSD model, applying the SENet model to introduce attention between channels. Figure 7 shows the SE block structure.

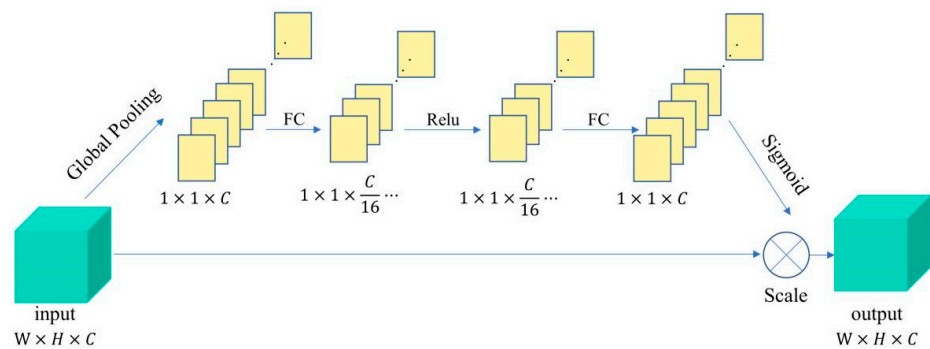

**Figure 7.** SE block structure.

In this paper, on the premise of converting the PSPNet backbone network into MobileNetv2, six feature maps of different scales are extracted, which can aggregate the information of different feature maps and ensure that the network can extract richer semantic information. Since the original network undergoes various convolution and compression operations, the original feature information is lost, and so this method is beneficial for the model to extract fracture texture features and edge information. The five different feature maps were first passed through the attention mechanism network SENet to increase the weight of important information; accordingly, the weight of unimportant information was suppressed before entering the Pyramid Scene Parsing module. Figure 8 shows the architecture of the proposed model.

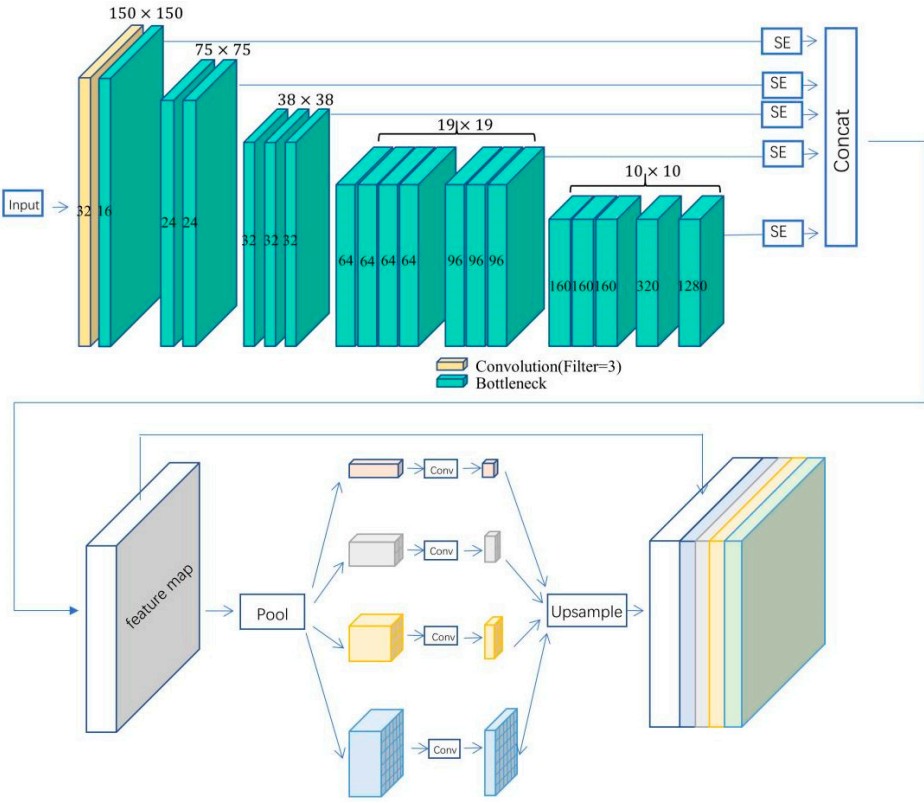

**Figure 8.** PSPNet architecture diagram with MobileNetv2 as backbone.

## 4. Data Preparation

### 4.1. Datasets Build

An FPSO dataset was collected from cracks of the upper module support pier. Figure 9 shows part of the FPSO dataset. As a total of 336 welds connecting the module support pier to the main deck were detected manually, where a total of 71 welds were reported to have cracks. The detection of these cracks was time-consuming and inefficient, so the collected photos were arranged, and 272 photos were selected as the dataset.

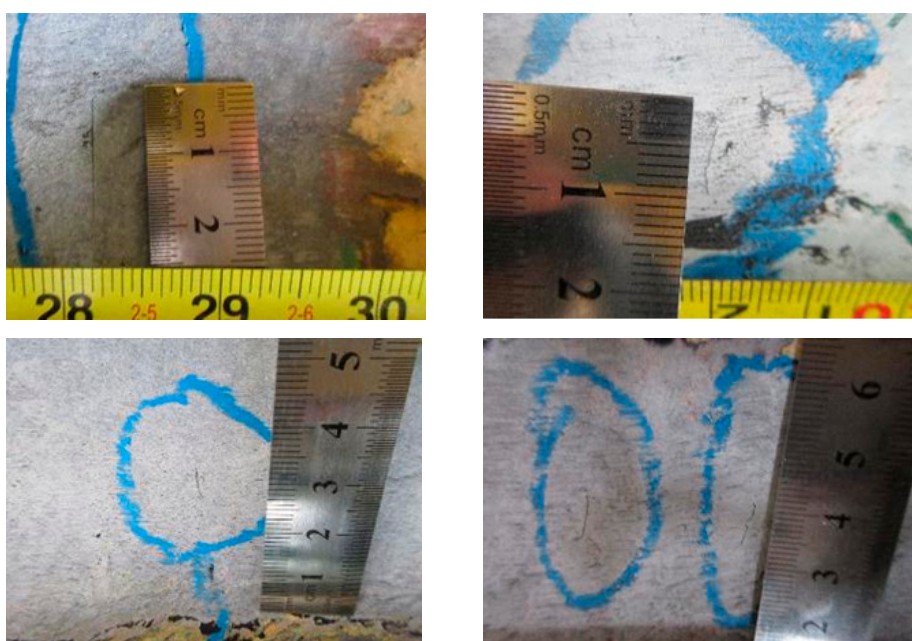

**Figure 9.** FPSO datasets.

Labelme is an open-source annotation tool mainly for semantic segmentation class datasets. Labelme can annotate image data in various forms. Labelme stores annotation information in JSON files. In this work, we used the software to create a semantic segmentation class annotation of the FPSO dataset. The crack data annotation is shown in Figure 10.

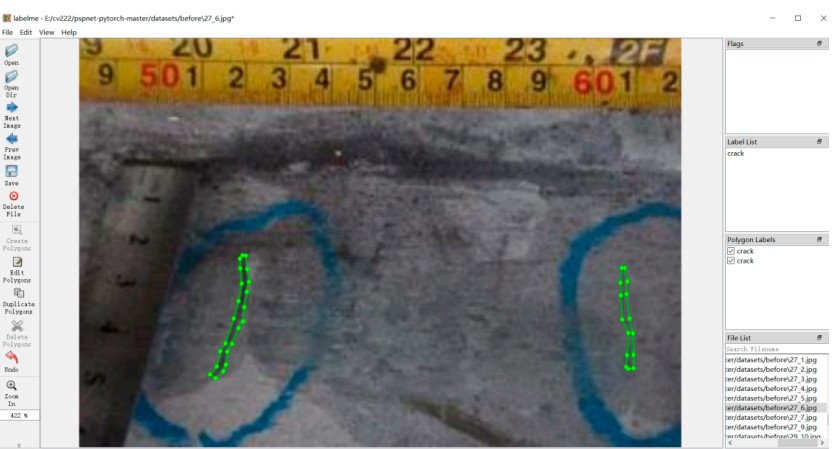

**Figure 10.** The crack data annotation.

*4.2. Data Enhancement*

To avoid issues associated with insufficient data and enhance the generalization capability of the model, methods of data enhancement to increase the image data were adopted. The specific methods include flipping, cropping, scaling, Gaussian noise, data loss, brightness adjustment, Gaussian blur, etc. The original image can usefully be flipped horizontally or vertically. It may be cropped in the following ways: random crop, target area crop, and split crop. The original image may also be rotated. Zoom enables the researcher to reduce or enlarge the original image. Gaussian noise adds pixel noise with a normal distribution to the image. Data loss removes image information of a predefined size in a randomly located area. Brightness adjustment tweaks the lighting of the original image. Gaussian blurring uses Gaussian kernel smoothing for each pixel of the image to obtain information about each nearby pixel.

In this step of our work, each image in the dataset was randomly selected for data enhancement by four of the above methods. One or more of the above methods was randomly selected to expand the data volume. Figure 11 presents details of the data enhancement effects.

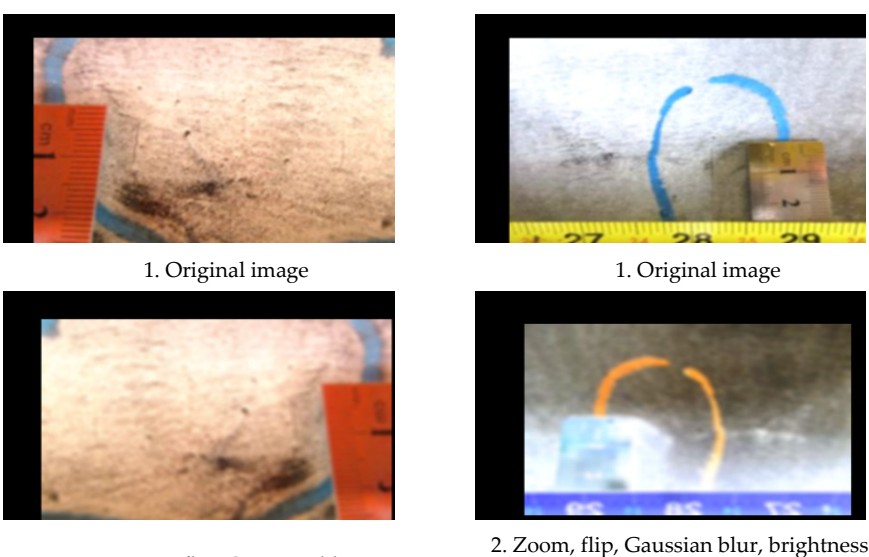

1. Original image       1. Original image

2. Zoom, flip, Gaussian blur       2. Zoom, flip, Gaussian blur, brightness adjustment

**Figure 11.** *Cont.*

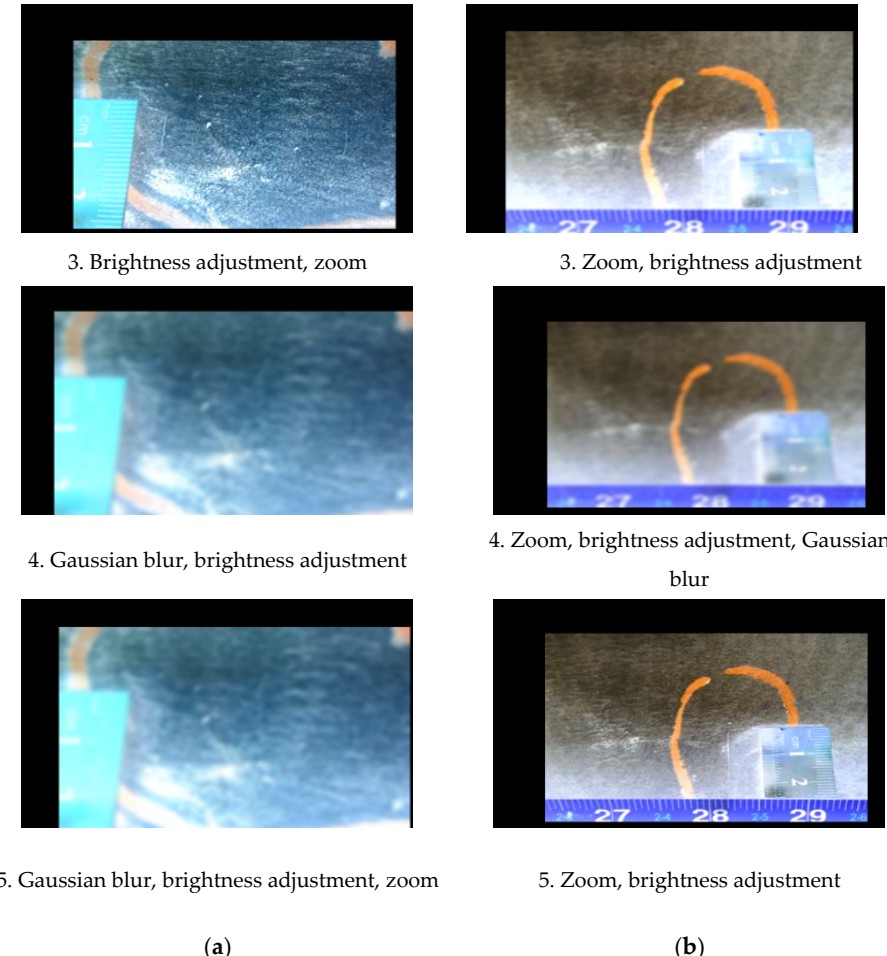

3. Brightness adjustment, zoom　　　　　　　3. Zoom, brightness adjustment

4. Gaussian blur, brightness adjustment　　　4. Zoom, brightness adjustment, Gaussian blur

5. Gaussian blur, brightness adjustment, zoom　　5. Zoom, brightness adjustment

(**a**)　　　　　　　　　　　　　　　　　　　　　(**b**)

**Figure 11.** Image enhancement example of two images: (**a**) Random Image Enhancement Combination 1; (**b**) Random Image Enhancement Combination 2.

## 5. Training

### 5.1. Loss Function

The loss function used in this paper consists of two parts: one is the classification error generated by classification using Cross-Entropy Loss [25], and the other is the segmentation error generated by pixel segmentation using Dice Loss [26]. Cross-Entropy Loss is used when the network applies the SoftMax function to classify pixels. In the case of binary classification, the model only needs to predict two cases, and the predicted probabilities are P and 1-P. The expression is as follows:

$$L = \frac{1}{N}\sum_i L_i = \frac{1}{N}\sum_i -[y_i \cdot \log(p_i) + (1 - y_i) \cdot \log(1 - p_i)] \tag{1}$$

where $y_i$ represents the label of sample *I*, with positive class 1 and negative class 0. *p* is the probability that sample *I* is positive. In the case of multiple categories, the dichotomies are extended as follows:

$$L = \frac{1}{N}\sum_i L_i = \frac{1}{N}\sum_i -\sum_{c=1}^{M} y_{ic} \log(p_{ic}) \tag{2}$$

where *M* refers to the number of categories, and $y_{ic}$ is the indicator variable (0 or 1). If the category is the same as sample *I*, it is 1, and if the category is different, it is 0, which refers to the predicted probability of sample *I* belonging to class C.

The similarity measure coefficient (dice) is used to calculate the similarity of the two samples, where the range is [0, 1]. The calculation formula of the similarity measure function is as follows:

$$dice = \frac{2|X \cap Y|}{|X|+|Y|} \tag{3}$$

where $X$ is the predicted result and $Y$ is the real result. The larger the result, the higher the coincidence degree between the predicted and real results. Therefore, the closer the dice coefficient is to 1, the better. If used as an error, the loss value should be as small as possible, so the loss of semantic segmentation should be set as Dice Loss = $1 -$ dice.

*5.2. Metrics*

The confusion matrix is another comprehensive measure of computer vision models, which includes the TP, FN, FP, and TN metrics. The confusion matrix is the basis for all other types of metrics. These metrics are obtained by mathematical operations on the data in the confusion matrix. The details of the confusion matrix are shown in Table 1.

**Table 1.** Confusion matrix.

| Category | | Predicted | |
|---|---|---|---|
| | | Positive | Negative |
| Real | Positive | TP | FN |
| | Negative | FP | TN |

The *recall* metric, shown in the formula below, consists of *TP* and *FN* and represents the proportion of positive samples that are both positive and predicted to be positive.

$$Recall = \frac{TP}{TP + FN} \tag{4}$$

*Precision*, shown in the formula below, is an indicator composed of *TP* and *FP*. Generally, a specified threshold is set to determine the confidence level of image classification to determine the *TP* and *FP* of image classification; this indicator represents the proportion of positive samples correctly predicted.

$$Precision = \frac{TP}{TP + FP} \tag{5}$$

The mean pixel accuracy and mean intersection over union are usually used as evaluation indexes in image segmentation. The higher the values of these indexes, the better the performance of the model.

Mean pixel accuracy: calculate the proportion of correctly classified pixels in each class, then calculate the average of this proportion in all classes:

$$mPA = \frac{1}{m+1} \sum_{i=0}^{m} \frac{p_{ii}}{\sum_{j=0}^{m} p_{ij}} \tag{6}$$

Mean intersection over union: calculate the ratio of the intersection and union of two sets, which are real and predicted in semantic segmentation problems. This ratio can be deformed to be the sum of true positives, true negatives, false negatives, and false negatives (unions). Calculate the IoU for each class, then take an average.

$$MIoU = \frac{1}{m+1} \sum_{i=0}^{m} \frac{p_{ii}}{\sum_{j=0}^{m} p_{ij} + \sum_{j=0}^{m} p_{ji} - p_{ii}} \tag{7}$$

## 5.3. Experimental Conditions

The division ratio of the training, validation, and test sets in this paper was 8:1:1. To prove the generalization of the model, the subsets consisting of training and validation sets, and the subset consisting of the test set, were enhanced separately. Specifically, the training set and verification set was composed of 1210 enhanced images produced from 242 of the original images. The training set then contained 1060 images and the verification set contained 150 images. The test set, meanwhile, was composed of 150 enhanced images produced from 30 of the original images.

Using the adaptive moment estimation (Adam) optimization algorithm, we found the learning rate was 0.001, $\beta 1$ was 0.9, and $\beta 2$ was 0.999 [27]. We set the batch size to 16. The model was trained for 350 epochs in total, and the training time was about 6 h. The experiment took about 0.33 s to process each image. Table 2 shows experimental conditions in this paper.

**Table 2.** Experimental conditions.

| Memory | 32 GB |
|---|---|
| GPU | GeForce RTX 2080 Ti |
| OS | Ubuntu 18.04 |
| Python | 3.6.8 |
| CUDA | 10.1 |
| Pytorch | 1.7.1 |

## 6. Experimental Results and Analysis

To verify the effectiveness of the proposed model, different algorithms were used for training. The prediction results of different models on the same test set are shown in Tables 3 and 4.

**Table 3.** Results of different models for training.

| Model | | Recall | Precision | PA | IOU |
|---|---|---|---|---|---|
| PSPNet | Background | 97.85 | 99.03 | 97.85 | 97.47 |
| | crack | 86.89 | 66.28 | 86.89 | 60.03 |
| MobileNet-PSPNet | Background | 98.19 | 98.83 | 98.19 | 97.91 |
| | crack | 87.17 | 70.15 | 87.17 | 64.04 |
| Improved model | Background | 98.58 | 99.52 | 98.58 | 98.29 |
| | crack | 90.39 | 74.06 | 90.39 | 68.47 |

**Table 4.** Performance comparison of the different models.

| Model | MioU | mPA | Accuracy |
|---|---|---|---|
| FCN | 62.64 | 78.51 | 89.75 |
| SegNet | 71.94 | 83.67 | 93.32 |
| DeepLabv3 | 79.11 | 91.08 | 97.96 |
| Unet | 75.29 | 88.21 | 86.14 |
| PSPNet | 78.75 | 92.37 | 97.87 |
| MobileNet-PSPNet | 80.98 | 92.68 | 98.01 |
| Improved model | 83.38 | 94.49 | 98.74 |

MobileNetv2 as the backbone showed an excellent performance, where a basic attention mechanism was introduced to enhance the performance of the model. This is because the background of the dataset in this paper was similar to the target object, and it was easy to recognize tiny cracks in the background during semantic segmentation. After the attention mechanism was introduced before feature extraction, the ability to identify cracks and extract contextual information was enhanced, as this is effective at recognizing tiny target objects. Table 3 and Figure 12 show the four performance indicators—recall,

precision, PA, and IoU—for assessment of the background and cracks with different models. It can be seen that the crack-detection performance of the improved model was improved by 3.5%, 7.78%, 3.5%, and 8.44%, respectively, compared to PSPNet, and 3.22%, 3.91%, 3.22%, and 4.43% versus Mobilev2-PSPNet.

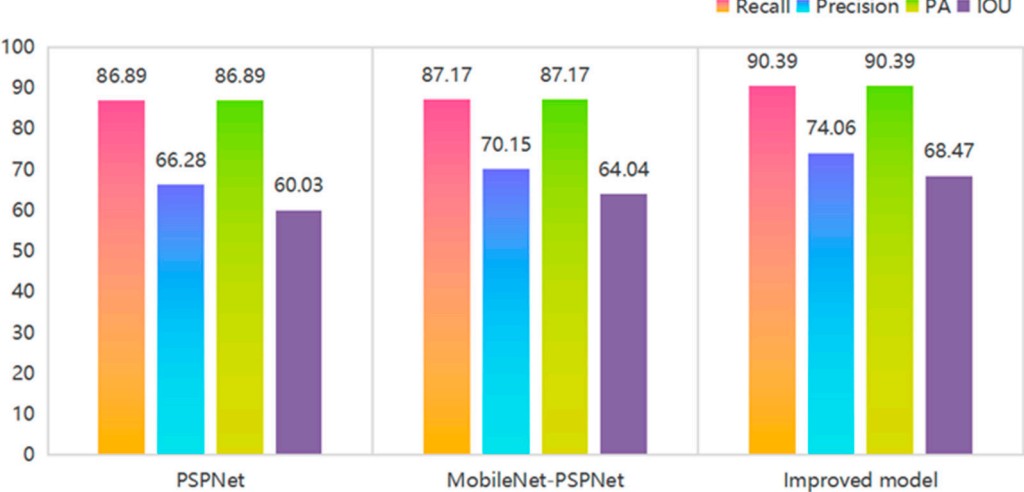

**Figure 12.** Result of training.

To further verify the effectiveness of the model, we compared it to the classic semantic segmentation FCN model and the most popular semantic segmentation model at present. It can be seen that the model proposed in this paper for FPSO cracks is practical. Table 4 compares the performance levels of PSPNet, Mobilev2-PSPNet, and the improved model. It can be seen that the MioU, PA, and accuracy of the improved model were 4.63%, 2.12%, and 0.87% higher than those of the PSPNet model, respectively. Compared to the Mobilev2-PSPNet model, it was 2.4%, 1.81%, and 0.73% better. Table 5 shows the number of parameters for different models. It can be seen from the table that using MobileNetv2 as the backbone greatly reduces the parameters of the model, and the model parameters after introducing the attention mechanism increase only slightly. As such, the performance metrics of the model are improved without a significant increase in model parameters, demonstrating the significance of our improved model.

**Table 5.** Number of parameters for different models.

| Model | No. of Parameters |
|---|---|
| PSPNet | 46,706,626 |
| MobileNet-PSPNet | 2,375,874 |
| Improved model | 2,388,058 |

As can be seen from the chart, we used MobileNetv2 as the model backbone, and the model performance was effectively promoted. Since the original network involves multiple convolutions and compressions, it is difficult to perform effective feature extraction. Moreover, cracks and the background can be easily confused due to the low resolution and the effects of light intensity, meaning the ResNet model may not be as good as MobileNetv2 at identifying small cracks. To solve this problem, the proposed model extracts different feature maps from MobileNetv2 and fuses them together after activating the attention mechanism. Experiments show that the method is effective. All the performance indexes are improved to some extent, which indicates that the model can improve the crack identification ability through different receptive fields after fusing different feature maps.

In Figures 13–15, (a) represents the loss curve of the PSPNet model, and (b) represents its MioU curve. As can be seen from the training curve, the loss gradually converges to

a stable value, while MioU gradually rises to a stable state. This demonstrates that the above three models are of practical significance. Figure 16 shows the results for crack identification by different means of semantic segmentation.

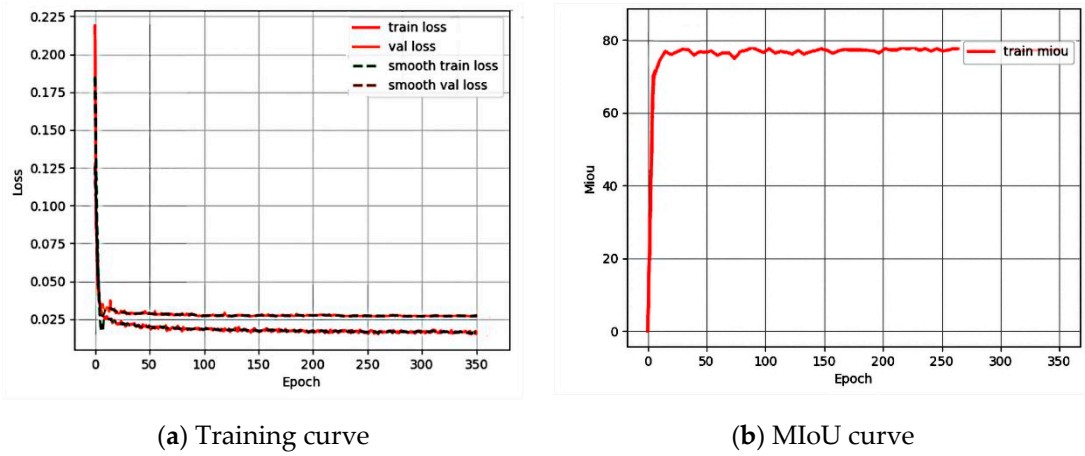

(**a**) Training curve
(**b**) MIoU curve

**Figure 13.** Training curve and MIoU curve of PSPNet model.

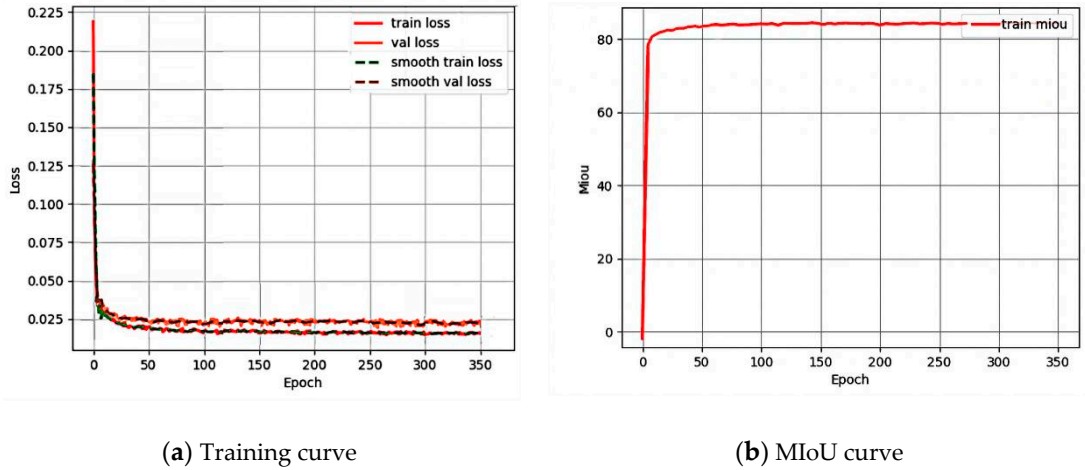

(**a**) Training curve
(**b**) MIoU curve

**Figure 14.** Training curve and MIoU curve of MobileNetv2-PSPNet model.

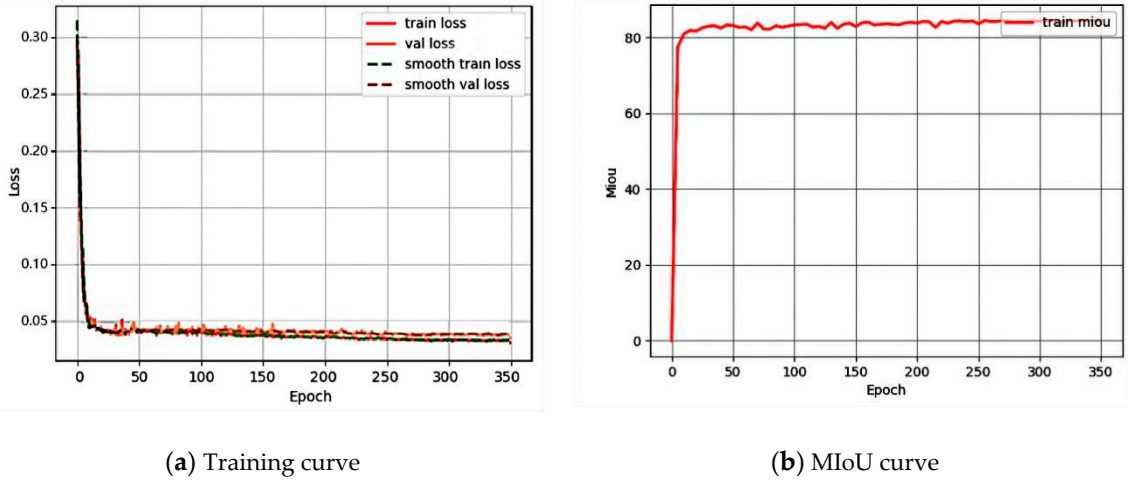

(**a**) Training curve
(**b**) MIoU curve

**Figure 15.** Training curve and MIoU curve of the improved model.

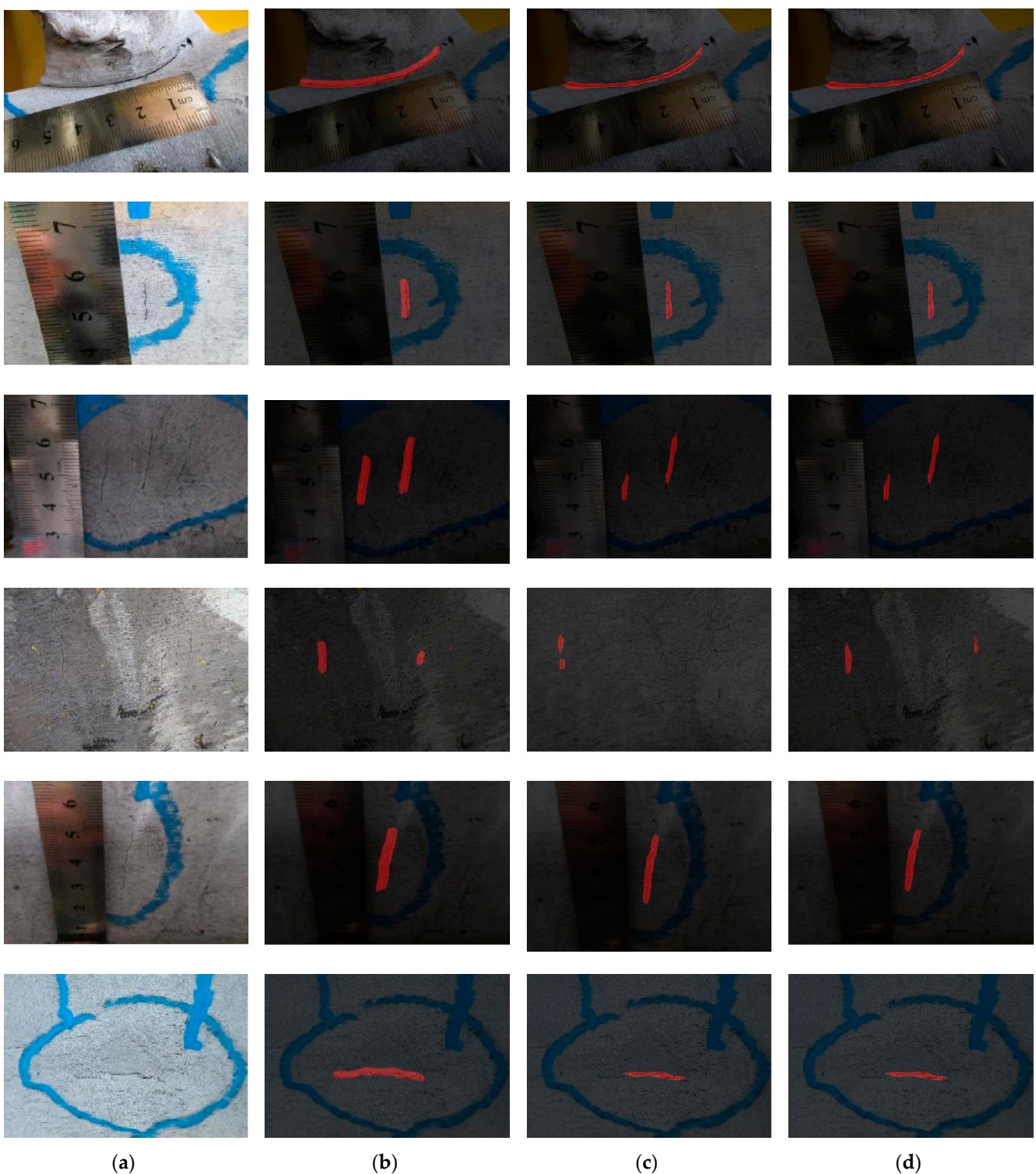

**Figure 16.** Results of crack identification by different semantic segmentation: (**a**) original image; (**b**) PSPNet; (**c**) MobileNetv2-PSPNet; (**d**) improved model.

## 7. Conclusions

For FPSO, real-time detection of cracks is an important guarantee of structural safety, and it is of great significance to know how to identify cracks using computer vision instead of manual detection and identification. In that context, the existing PSPNet model was improved in this paper. In a case where the original model performed excellently, the

main trunk was changed to MobileNet and attention mechanism SENet was introduced. Training was carried out on the FPSO dataset we made, and the performance of the original model was improved. The important evaluation index values of mPA and MIoU semantic segmentation models were improved obviously, and an improved model for FPSO crack identification was obtained. The improved model has engineering significance for FPSO crack identification. The proposed method of multi-feature graph fusion enhances the feature extraction ability of the model for fine cracks and effectively integrates contextual information. In our research presented here, the MIoU and mPA of the improved model reached 83.38% and 94.49%, respectively, indicating the effectiveness of the model we propose.

**Author Contributions:** Conceptualization, Z.J.; methodology, X.S.; software, X.S and G.M.; validation, Z.J., X.S. and C.Q.; formal analysis, T.D.; investigation, L.R.; resources, G.M.; data curation, X.S.; writing—original draft preparation, Z.J.; writing—review and editing, C.Q.; visualization, G.M.; supervision, T.D.; project administration, L.R.; funding acquisition, Z.J. and C.Q. All authors have read and agreed to the published version of the manuscript.

**Funding:** This research was funded by National Key Research and Development Program of China grant number 2019YFB1504303, the National Natural Science Foundation of China grant number 52078100, Anhui international joint research center of data diagnosis and smart maintenance on bridge structures grant number 2022AHGHYB03 and the Fundamental Research Funds for the Central Universities grant number DUT22JC19.

**Institutional Review Board Statement:** Not applicable.

**Informed Consent Statement:** Not applicable.

**Data Availability Statement:** No data were used to support this study.

**Conflicts of Interest:** The authors declare no conflict of interest.

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
