# Peer review of "Image-Based Crack Detection Method for FPSO Module Support"

_buildings, doi:10.3390/buildings12081147_

Round 1

Reviewer 1 Report

The manuscript has been modified according to the reviewer's comments.

Author Response

We thank the reviewer for reviewing our manuscript and offering positive comments.

Reviewer 2 Report

The paper is suitable for publication. 

Author Response

(The authors gave the same response as above.)

Reviewer 3 Report

The authors have made major revisions according to the previous comments. However, there are still some points ignored. Please tread them thoughtfully.

Once they are done, this paper is recommended for publication

1. The English writing should be significantly improved since the authors did not treat it seriously. I have rewritten the abstract for the authors. However, as I have stated before, English writing of the other part is inadequate in its current form.

Floating Production Storage and Offloading (FPSO) is essential offshore equipment for developing offshore oil and gas. Due to the complex sea conditions, FPSOs will be subjected to long-term alternate loads under some circumstances. Thus, it is inevitable that small cracks occur in the upper part of the module pier. Those cracks may influence the structure's safety evaluation. Therefore, this paper proposes a method for the FPSO module to support crack identification based on the PSPNet model. The main idea is to introduce an attention mechanism into the model with Mobilenetv2 as the backbone of the PSPNet, which can fuse multiple feature maps and increase context information. The detail feature loss caused by multiple convolution and compression in the original model was solved by applying the proposed method. Besides, the attention mechanism is introduced to enhance the extraction of adequate information and suppress invalid information. The mPA value and MIoU value of the improved model increased by 2.4% and 1.8%, respectively, through verification on FPSO datasets.

2. As stated before, the authors should clarify how the dataset is divided. Besides, the hyper parameters should also be given..

The authors explained them in lines 314 to 322. However, the English writing should be improved. It took me 10 minutes to understand what the authors wanted to explain.

Another thing is the term “two subsets” in line 314. What does it mean?

3. The model weight (how many parameters each network should optimize), required training epochs, and 1-Batch (or one epoch) and inference time of each model should also be posted in a Table and a chart format.

4. Again, the figure quality should be improved.

Please improve the quality and font size to be more visible (Figure13-15).

Please improve the graphic quality (resolution, Figure 3-8).

Author Response

We would like to resubmit the manuscript titled "FPSO Module Supported Image-Based Crack Detection Method", by Su Xin, Jia Ziguang, Ma Guangda, Qu Chunxu, Dai Tongtong, and Ren Liang under manuscript ID building- 1830174.

We thank the reviewers for the time and effort that they have put into reviewing the previous version of the manuscript. At the same time, we have polished the full text through MDPI. Their suggestions have enabled us to improve our work.

We deeply appreciate your consideration of our manuscript, and we look forward to receiving comments from the reviewers. If you have any queries, please don't hesitate to contact me at the address below.

Thank you and best regards.

Yours sincerely,

Xin, Su

Chunxu Qu, Ph.D.

Associate Professor

Director, Institute of Structural Diagnosis and Rehabilitation (ISDR)   

School of Civil Engineering, Dalian University of Technology

500 3rd Laboratory building, 2 Linggong Road, Ganjingzi District

Dalian, Liaoning Province, 116024, China

Phone: +86-15140500187

E-Mail: quchunxu@dlut.edu.cn

Round 2

Reviewer 3 Report

I have no issue with this manuscript, and it is recommended to be published its current form

This manuscript is a resubmission of an earlier submission. The following is a list of the peer review reports and author responses from that submission.

Round 1

Reviewer 1 Report

In this manuscript, a PSPNet model-based method for crack identification of floating production storage and offloading (FPSO) module supports is proposed. The Mobilenetv2 backbone is applied to improve the performance of the original PSPNet. The manuscript presents an interesting research and falls within the scope of the journal of Building. However, this manuscript is not well written and the innovation is not significant. The detailed comments are as follows:

(1)     The main idea of this method is to replace the original backbone of PSPNet with Mobilenetv2. However, similar methods can be found in some published paper, e.g. “Liu B Y, Fan K J, Su W H, et al. Two-Stage Convolutional Neural Networks for Diagnosing the Severity of Alternaria Leaf Blotch Disease of the Apple Tree[J]. Remote Sensing, 2022, 14(11): 2519.”

(2)     In data preparation, 272 photos were selected as the dataset. However, the sizes of the training set after data enhancement is unknown. The information of the validation set and the testing set is not mentioned.

(3)     In this manuscript, only the training results of the proposed model are presented. It is difficult to determine whether overfitting has occurred. However, we should pay more attention to its prediction accuracy on the validation set and the testing set, which reflects the generalization ability of the model.

(4)     There are many text errors in the manuscript, some are as follows:

1)      Abbreviations need to be defined before they can be used. But the use of the abbreviations in this manuscript is irregular, e.g. “Production Storage and Offloading (FPSO)”.

2)      It is recommended not to use too long sentences, it is not easy to understand. e.g. “Different depth of features is based on the input through the different scales of pooling operation, then through 1x1 convolution layer will feature dimension reduction for 1/4 of the original, will finally these pyramid features directly on the sampling to the same size with the input characteristics, and then merged and input characteristics to get the final output features.” In page 6.

3)      In page 2, the commas after “How to effectively detect the crack is a crucial problem” should be replaced by periods.

4)      In page 2, the word “a” in “Neural networks have undergone many innovations, and a crack recognition method based on deep learning has been widely used.” is inappropriate use.

5)      In page 3, a space is absent after the sentence “Since then, CNN has entered the industrial world.”

6)      In page 3, in the sentence “In page 3, a space is absent after the sentence “Since then, CNN has entered the industrial world.”, the font of the word “then” is wrong.

7)      There should be a comma before “respectively”.

8)      In page 3 and 4, The first letter of the word “proposed” should be lowercase. e.g. “Zheng et al. Proposed an algorithm based on SegNet and separable convolution of bottleneck depth with residuals, which can be used for high-precision light bridge concrete crack detection”.

9)      In page 4, two spaces are absent in “conv4_xandconv5_x”.

10)  Please check the full paper carefully.

Reviewer 2 Report

The language of the manuscript is vague and need rewriting the whole manuscript. This manuscript does not provide standard novelty for a research paper, and it has not been found suitable for publication in the Buildings journal. Some figures are not clear, and the text is not readable. This is a rejected paper.

Reviewer 3 Report

please check the attachment 
